# Microbial Assemblages Associated with the Soil-Root Continuum of an Endangered Plant, *Helianthemum songaricum* Schrenk

Daolong Xu,[a] Xiaowen Yu,[b] Jin Chen,[c] Haijing Liu,[a] Yaxin Zheng,[a] Hanqing Qu,[a] (ID) Yuying Bao[a]

[a]Inner Mongolia University, School of Life Sciences, Ministry of Education, Key Laboratory of Forage and Endemic Crop Biotechnology, Hohhot, People's Republic of China

[b]Inner Mongolia Autonomous Region Environmental Monitoring Station, Hohhot, People's Republic of China

[c]National Engineering Laboratory of Crop Stress Resistance Breeding, Anhui Agricultural University, Hefei, People's Republic of China

**ABSTRACT** The microbial network of the soil-root continuum plays a key role in plant growth. To date, limited information is available about the microbial assemblages in the rhizosphere and endosphere of endangered plants. We suspect that unknown microorganisms in roots and soil play an important role in the survival strategies of endangered plants. To address this research gap, we investigated the diversity and composition of the microbial communities of the soil-root continuum of the endangered shrub *Helianthemum songaricum* and observed that the microbial communities and structures of the rhizosphere and endosphere samples were distinguishable. The dominant rhizosphere bacteria were *Actinobacteria* (36.98%) and *Acidobacteria* (18.15%), whereas most endophytes were *Alphaproteobacteria* (23.17%) as well as *Actinobacteria* (29.94%). The relative abundance of rhizosphere bacteria was higher than that in endosphere samples. Fungal rhizosphere and endophyte samples had approximately equal abundances of the *Sordariomycetes* (23%), while the *Pezizomycetes* were more abundant in the soil (31.95%) than in the roots (5.70%). The phylogenetic relationships of the abundances of microbes in root and soil samples also showed that the most abundant bacterial and fungal reads tended to be dominant in either the soil or root samples but not both. Additionally, Pearson correlation heatmap analysis showed that the diversity and composition of soil bacteria and fungi were closely related to pH, total nitrogen, total phosphorus, and organic matter, of which pH and organic matter were the main drivers. These results clarify the different patterns of microbial communities of the soil-root continuum, in support of the better conservation and utilization of endangered desert plants in Inner Mongolia.

**IMPORTANCE** Microbial assemblages play significant roles in plant survival, health, and ecological services. The symbiosis between soil microorganisms and these plants and their interactions with soil factors are important features of the adaptation of desert plants to an arid and barren environment. Therefore, the profound study of the microbial diversity of rare desert plants can provide important data to support the protection and utilization of rare desert plants. Accordingly, in this study, high-throughput sequencing technology was applied to study the microbial diversity in plant roots and rhizosphere soils. We expect that research on the relationship between soil and root microbial diversity and the environment will improve the survival of endangered plants in this environment. In summary, this study is the first to study the microbial diversity and community structure of *Helianthemum songaricum* Schrenk and compare the diversity and composition of the root and soil microbiomes.

**KEYWORDS** *Helianthemum songaricum*, high-throughput sequencing, endangered plants, soil-root continuum microbial community, diversity

**Ad Hoc Peer Reviewers** (ID) Gaosen Zhang, Key Laboratory of Desert and Desertification, Northwest Institute of Eco-Environment and Resources, Chinese Academy of Sciences, Lanzhou, China; Key Laboratory of Extreme Environmental Microbial Resources and Engineering, Gansu Providence, Lanzhou, Gansu, China; Madhusmita Borah, Assam Agricultural University

Address correspondence to Yuying Bao, ndbyy@imu.edu.cn.

The authors declare no conflict of interest.

Soil has high-abundance and diverse microbial communities that have multiple functions and play key roles in ecosystem functioning (1, 2). Plants, microorganisms, and the soil have formed mutually beneficial relationships that can help plants adapt to a variety of environmental changes (3, 4). In this relationship, microorganisms of the soil-root continuum can stimulate the growth of the plants by providing antibiotics, plant growth, nutrients, and hormones that improve plant stress resistance and avoid infection by pathogens (5, 6). Furthermore, the exudates and deposits from the roots of plants, such as a broad range of carbohydrates and amino acids, can help in the propagation of certain microorganisms (7). Consequently, discovering the assembly of the microorganism community of the soil-root continuum can increase the adaptation of plants to extreme environments.

Currently, the composition of microbial communities of endangered plants, including *Sarcozygium xanthoxylon*, *Tetraena mongolica*, and *Nitraria tangutorum* Bobr, has been studied via high-throughput sequencing technology (8). Previous studies demonstrated that the structure and composition of microbial communities in the roots and soil were different, and the microbial species in the soil were more varied than those in the endosphere (9). However, the microbiome of the soil-root continuum of a typical endangered plant, *Helianthemum songaricum* Schrenk, has not been studied previously; therefore, little is known about whether or not soil microbial variation is correlated with succession in these endangered plants. To determine the differences between the bacterial and fungal communities of *Helianthemum songaricum* Schrenk growing in its natural habitat, the microbial community structure in the soil-root continuum was determined.

It has been known that the microbial community of the soil-root continuum exerts a considerable beneficial influence on the growth of plants (10). Such a complicated microbial community can enhance the tolerance of plants to stressful environments, including improvements in the capacity to resist disease and stress and increased plant growth rates (11, 12). As reported previously, the microbial community is affected by root exudates and the external environment and thus exhibits complex diversity. In contrast, the root microbial groups show highly specific and relatively stable structural features (13). The microbes of the soil-root continuum, including bacterial and fungal communities, form different microbial networks to respond to changes in the external environment. In addition, these microbial network changes reflect changes in the interactions between microorganisms, which can promote the soil environment and plant growth (14). Previous studies reported that external environmental disturbances such as drying and disease can reduce the network stability of microorganisms in the soil, which is related to the vegetation composition and soil moisture (15). However, in natural states, the microbial groups of the soil-root continuum and their differences *in situ* remain unclear. Moreover, various environmental factors (e.g., soil total nitrogen [TN] content, pH value, and total phosphorus [TP] content) may impact the differences in the responses of the microbiome communities of the soil-root continuum (16).

To address these issues, we analyzed the fungal and bacterial communities of *Helianthemum songaricum* Schrenk growing in Ordos City, Qipanjing, People's Republic of China, via high-throughput sequencing technology, enabling direct comparisons between these microbial communities. The main purposes were to reveal the variations in soil-root microbial communities, elucidate the environmental factors that influence the root and soil microbial communities, and provide a theoretical reference for the better conservation and utilization of endangered desert plants in Inner Mongolia.

## RESULTS

**Structure and diversity of the microbial communities.** *Helianthemum songaricum* Schrenk root and soil samples were examined by high-throughput sequencing. After quality control and trimming, totals of 186,074 (approximately 821.392 bp) and 87,893 (approximately 527.236 bp) gene sequences of bacterial 16S rRNA and fungal internal transcribed spacer (ITS) sequences were produced, respectively (see Table S2 in the

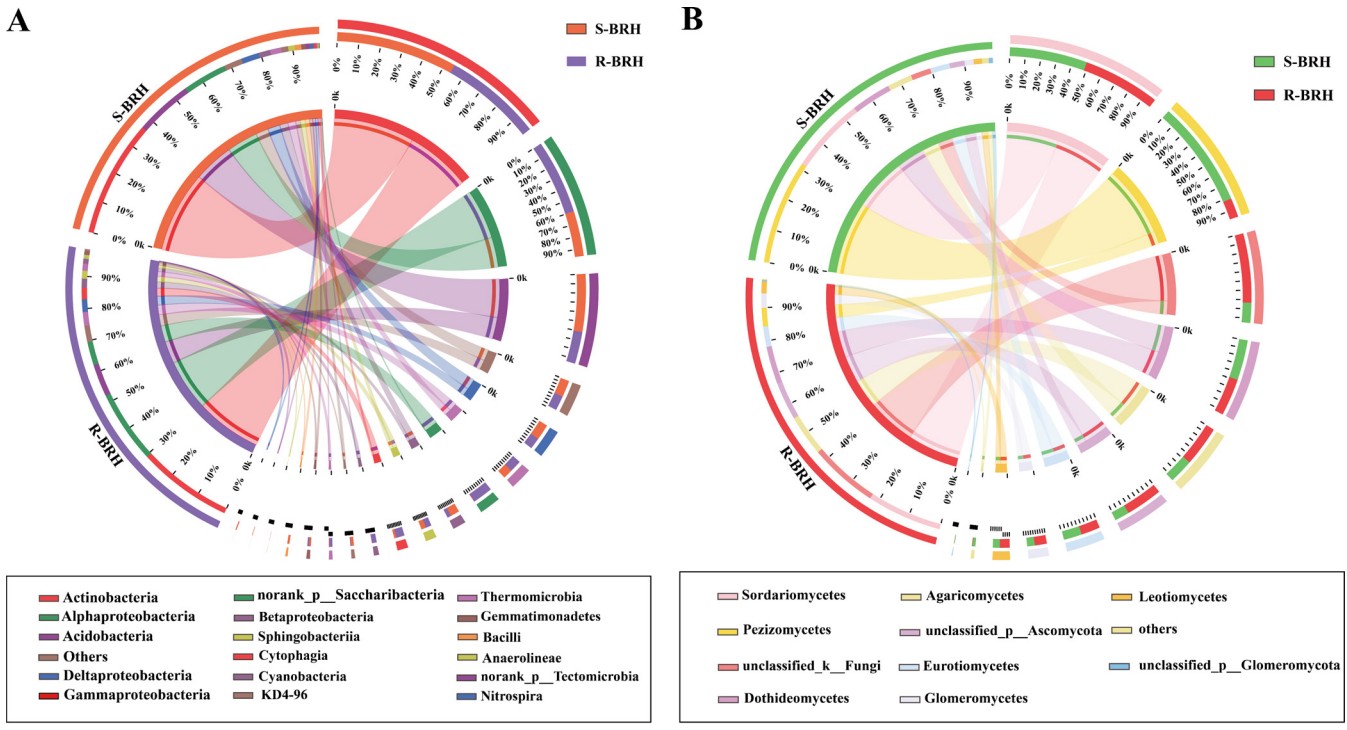

**FIG 1** Distributions of bacterial (A) and fungal (B) communities in each sample at the class level. The width of the bar for each phylum indicates the class relative abundance for that sample.

supplemental material). Subsequently, 3,050 operational taxonomic units (OTUs) (cutoff of 97.0%) were found by clustering and alignment, with 2,432 OTUs of the bacterial 16S rRNA gene (1,214 OTUs were shared) (Fig. S1a) and 618 OTUs of the fungal ITS (183 OTUs were shared) (Fig. S1b). All of the samples' rarefaction curves had achieved a plateau (Fig. S2), indicating that the sequencing depth was adequate and that all of the data were credible and authentic.

In this study, 18 bacterial classes were identified in S-BRH and R-BRH, and the dominant classes (relative abundance of >1% in at least one sample) comprised *Actinobacteria* (36.98% and 29.94%, respectively), *Acidobacteria* (18.15% and 10.12%, respectively), *Alphaproteobacteria* (14.06% and 23.17%, respectively), *Deltaproteobacteria* (5.62% and 3.99%, respectively), *Betaproteobacteria* (3.75% and 1.58%, respectively), *Sphingobacteriia* (2.12% and 2.527%, respectively), *Cytophagia* (0.97% and 3.65%, respectively), *Gemmatimonadetes* (1.38% and 1.18%, respectively), *Thermomicrobia* (0.36% and 2.20%, respectively), *Cyanobacteria* (0.15% and 2.83%, respectively), and *Anaerolineae* (0.36% and 1.20%, respectively) (Fig. S3a and b). Nineteen fungal classes were identified in S-BRH and R-BRH, and the dominant classes included *Pezizomycetes* (31.95% and 5.70%, respectively), *Sordariomycetes* (23.02% and 22.91%, respectively), *Dothideomycetes* (12.21% and 11.43%, respectively), *Agaricomycetes* (7.55% and 12.46%, respectively), *Eurotiomycetes* (5.94% and 6.18%, respectively), and *Leotiomycetes* (2.59% and 3.66%, respectively) (Fig. S3c and d). The predominant bacterial and fungal components of samples from the rhizosphere soil and roots were different (Fig. 1). We found that the bacteria in S-BRH and R-BRH came mainly from the three most dominant classes: *Actinobacteria*, *Acidobacteria*, and *Alphaproteobacteria*. *Actinobacteria* were present at the highest percentages in S-BRH and R-BRH. The abundance of *Alphaproteobacteria* in R-BRH was significantly higher than that in S-BRH (Fig. 1A). The fungi in the rhizosphere soil and roots belonged to the following broad groups: *Dothideomycetes*, *Pezizomycetes*, unclassified_k_Fungi (unclassified fungal flora), and *Sordariomycetes*. It is quite clear that the fungal communities in the rhizosphere roots and soils are very different from each other. For example, in the soil, the abundance of *Pezizomycetes* in the rhizosphere was higher than that in R-BRH (Fig. 1B).

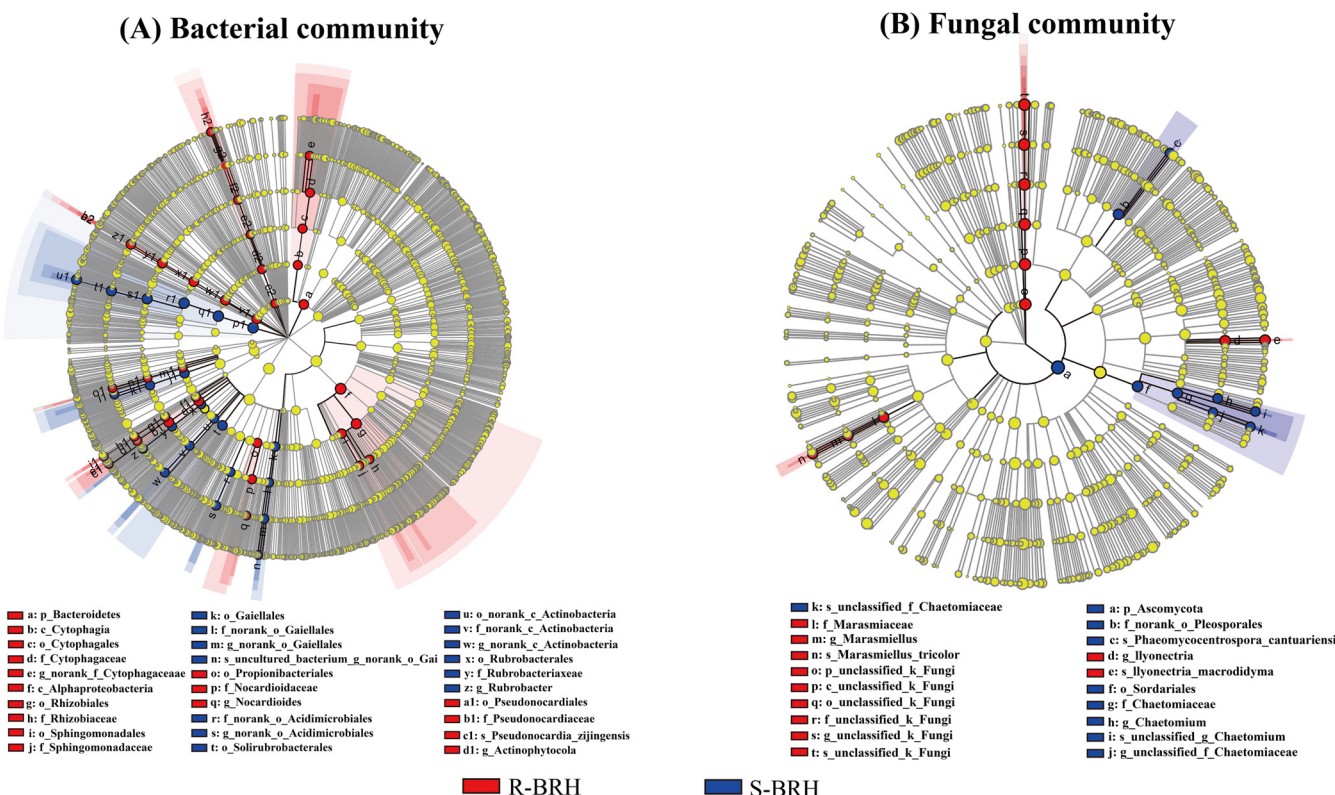

**(A) Bacterial community**

**(B) Fungal community**

a: p_Bacteroidetes
b: c_Cytophagia
c: o_Cytophagales
d: f_Cytophagaceae
e: g_norank_f_Cytophagaceae
f: c_Alphaproteobacteria
g: o_Rhizobiales
h: f_Rhizobiaceae
i: o_Sphingomonadales
j: f_Sphingomonadaceae

k: o_Gaiellales
l: f_norank_o_Gaiellales
m: g_norank_o_Gaiellales
n: s_uncultured_bacterium_g_norank_o_Gai
o: o_Propionibacteriales
p: f_Nocardioidaceae
q: g_Nocardioides
r: f_norank_o_Acidimicrobiales
s: g_norank_o_Acidimicrobiales
t: o_Solirubrobacterales

u: o_norank_c_Actinobacteria
v: f_norank_c_Actinobacteria
w: g_norank_c_Actinobacteria
x: o_Rubrobacterales
y: f_Rubrobacteriaxeae
z: g_Rubrobacter
a1: o_Pseudonocardiales
b1: f_Pseudonocardiaceae
c1: s_Pseudonocardia_zijingensis
d1: g_Actinophytocola

k: s_unclassified_f_Chaetomiaceae
l: f_Marasmiaceae
m: g_Marasmiellus
n: s_Marasmiellus_tricolor
o: p_unclassified_k_Fungi
p: c_unclassified_k_Fungi
q: o_unclassified_k_Fungi
r: f_unclassified_k_Fungi
s: g_unclassified_k_Fungi
t: s_unclassified_k_Fungi

a: p_Ascomycota
b: f_norank_o_Pleosporales
c: s_Phaeomycocentrospora_cantuariensis
d: g_llyonectria
e: s_llyonectria_macrodidyma
f: o_Sordariales
g: f_Chaetomiaceae
h: g_Chaetomium
i: s_unclassified_g_Chaetomium
j: g_unclassified_f_Chaetomiaceae

R-BRH          S-BRH

**FIG 2** LDA effect size (LEfSe) of the bacterial (A) and fungal (B) communities with an LDA score of >4.0 ($P < 0.05$). Microbial lineages and plant compartment associations are displayed by cladogram phylogenetic distributions. Phylogenetic levels from phylum to species are represented by circles. Different-colored nodes represent microbial groups that are significantly enriched in the corresponding groups and have a significant influence on the differences between groups. The light-yellow nodes represent microbial groups that have no significant difference between groups or have no significant effect on the differences between groups.

**Biomarkers identification by the LEfSe method.** The linear discriminant analysis (LDA) effect size (LEfSe) method was used in this study to measure the abundance effect of core microbiomes with differential effects and to identify core microbiomes that differed significantly (biomarkers) between R-BRH and S-BRH (Fig. 2). The identified biomarkers exhibited a significant amount of fluctuation in the relative abundances of their core species, which was followed by major shifts as a result of environmental variations. The LEfSe study revealed that compared with the fungal community (which consisted of 20 clades, 3 genera and species, 2 orders and families, and 1 phylum and class), the bacterial communities demonstrated a large degree of sensitivity to the environment of the roots (60 clades, 12 orders, 9 families, 7 genera, 5 classes, 4 phyla, and 3 species). Moreover, the LDA results revealed 38 and 22 phylotypes for the microbiota of the R-BRH and S-BRH bacterial samples, respectively. For fungi, 11 and 9 phylotypes of fungi were detected in the R-BRH and S-BRH microbiota samples, respectively (Fig. 3). *Firmicutes* and *Bacillales* with an LDA score of >5.0 were highly significant in the bacterial taxon category in the S-BRH samples, signifying that the microbial composition in the root system is different from the microbial composition in the soil, with the bacterial community having more biomarkers than the fungal community in extremely arid and barren environments.

Some studies have also shown that the communities of the rhizosphere have more root environmental sensitivity and biomarkers than the soil communities of the rhizosphere for fungi as well as bacteria (11 and 9 biomarkers for fungi and 38 and 22 biomarkers for bacteria, respectively). Moreover, in the bacterial communities, the proportions of *Alphaproteobacteria*, *Cytophagia*, *Saccharibacteria*, *Cyanobacteria*, *Actinophytocola*, and *Streptomycetales* showed significant increases in the compartment of R-BRH. Similarly, the *Firmicutes*, *Actinobacteria*, *Bacillales*, and *Rubrobacterales* showed significant increases in the compartment of S-BRH. Furthermore, in the fungal communities, the proportions of

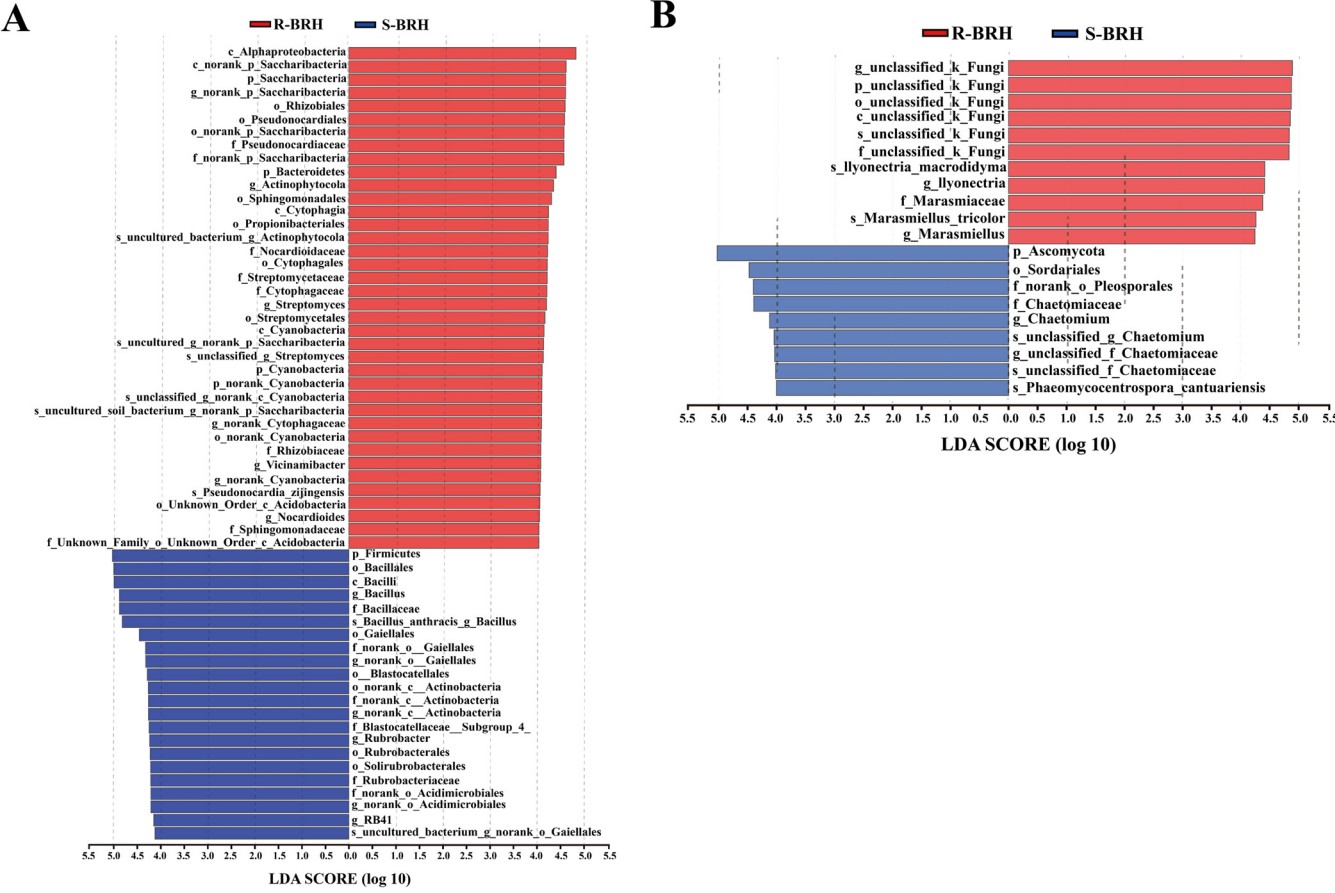

**FIG 3** LDA bar graphs indicating the communities of bacteria (A) and fungi (B) with an LDA score of >4.0.

*Macrodidyma*, *Ilyonectria*, *Tricolor*, and *Marasmiellus* were also significantly increased in the compartment of R-BRH, while the proportions of *Ascomycota*, *Sordariales*, *Pleosporales*, *Chaetomiaceae*, and *Chaetomium* were expressively increased in the compartment of S-BRH. Furthermore, the most significant fungal taxa in the sample of S-BRH were *Ascomycota* (LDA score of >5.0).

**Bacterial and fungal cooccurrence networks.** To illustrate the interactions between microbial species and the environment as well as the species correlations at each categorization level under a given environmental situation, cooccurrence and species correlation network analyses (Fig. 4) were used at each level of classification. A class is represented by the node in the network, while a link represents a cooccurrence connection between two linked nodes. The average shortest route lengths (the shortest path between two nodes) for the fungal and bacterial networks were 1.941 and 2.075, respectively. It was found that in S-BRH and R-BRH, 10 and 20 of the fungal and bacterial species were associated with one another, respectively. In extremely arid and barren environments, the communities of bacteria had increased complexity in a connected network. Furthermore, the nodes in a network of fungi were grouped into 17, while those in the bacterial network were grouped into 29 (Fig. 4A and B).

Further research revealed that the correlated bacterial communities in S-BRH and R-BRH included *Betaproteobacteria*, *Gammaproteobacteria*, *Cyanobacteria*, *Phycisphaerae*, *Acidobacteria*, *Alphaproteobacteria*, KD4-96, *Actinobacteria*, *Planctomycetacia*, *Cytophagia*, *Chloroflexia*, *Thermomicrobia*, *Sphingobacteriia*, *Gemmatimonadetes*, Gitt-GS-136, TK10, *Deltaproteobacteria*, and *Opitutae* (Fig. 4C). Simultaneously, the correlated fungal communities in S-BRH and R-BRH included *Dothideomycetes*, *Eurotiomycetes*, *Leotiomycetes*, *Pezizomycetes*, *Sordariomycetes*, *Ascomycota*, and *Agaricomycetes*. In general, bacterial networks were larger, had more nodes and linkages, and were more linked and complicated

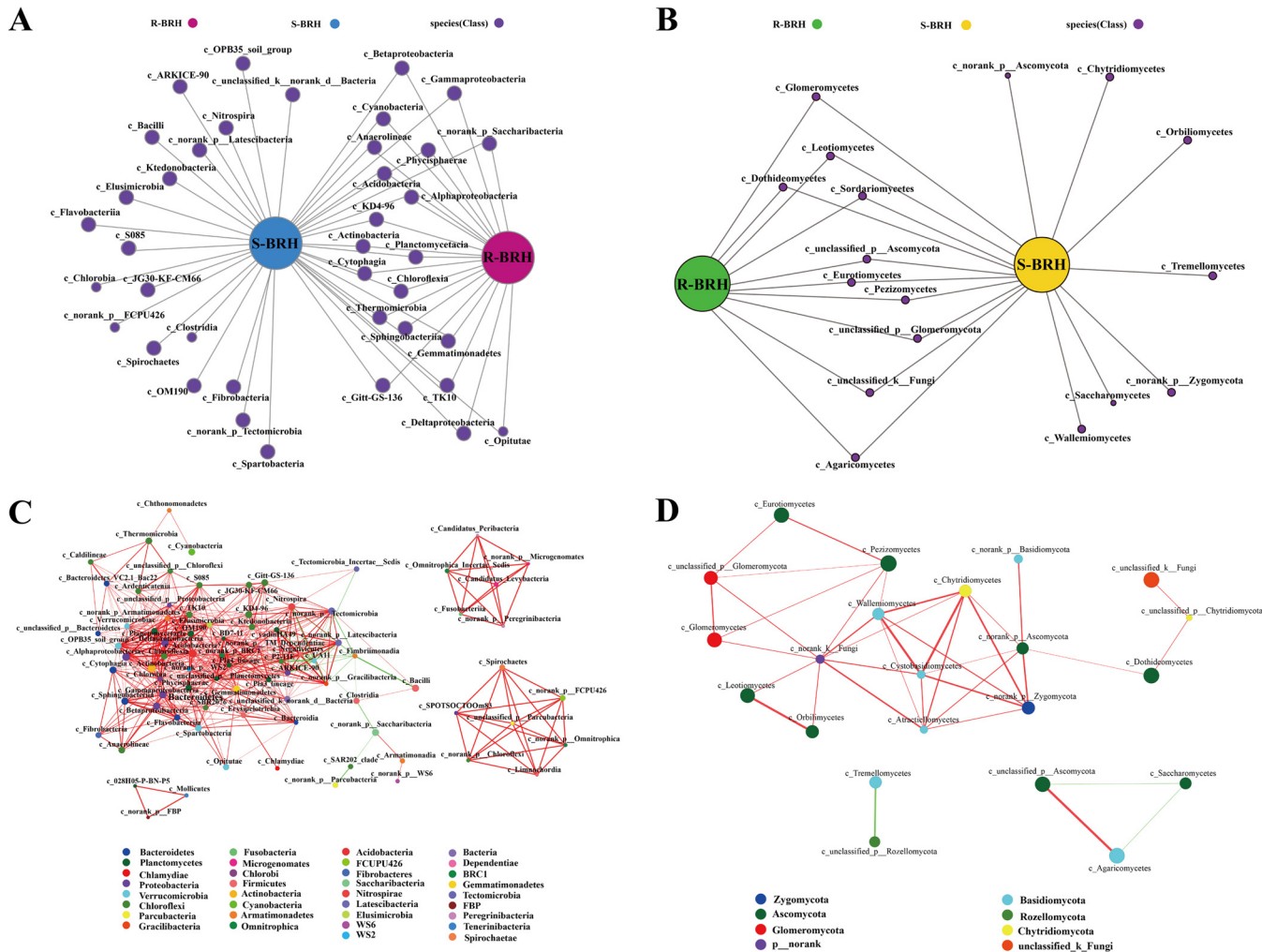

**FIG 4** Species correlation and cooccurrence network analyses at the class level between the *Helianthemum songaricum* Schrenk soil and root communities of bacteria (A and C) and fungi (B and D). The cooccurrence network diagram visually displays the cooccurrence relationships of species in different samples. The nodes in the network represent sample nodes or species nodes, and the lines between the sample nodes and species nodes represent the species contained in the samples. Species with an abundance (number of sequences) of >50 are displayed by default.

than fungal networks. Moreover, the *Fimbriimonadia* were negatively and significantly associated with nine categories, with a *P* value of <0.05, including *Bacilli*, *Clostridia*, *Tectomicrobia*, vadinHA49, UA11, *Negativicutes*, *Nitrospira*, ARKICE-90, and *Gracilibacteria*. The *Actinobacteria*, *Alphaproteobacteria*, *Deltaproteobacteria*, *Acidobacteria*, *Deltaproteobacteria*, and *Gammaproteobacteria* had higher species richness and presented advanced bacterial connections as well, while in the fungal group, the *Tremellomycetes* were negative but had a significant association with *Rozellomycota*. The *Saccharomycetes* were negatively correlated with the *Ascomycota* and *Agaricomycetes*. Additionally, the *Pezizomycetes* and *Chytridiomycetes* showed significant and positive correlations (*P* < 0.05) with other fungal groups (Fig. 4D).

**Phylogenetic analysis.** In parallel, a phylogenetic analysis was performed using MEGA with 1,000 replicates (maximum likelihood [ML] method), and evolutionary trees of the top 50 genera were drawn using the R language. In this study, the phylogenetic trees showed high credibleness: most of the clusters had a high bootstrap value, i.e., 60, as shown in Fig. 5. The soil and root microbiomes in the phylogenetic analysis were categorized into 31 fungal and 50 bacterial phyla. The phylogenetic analysis also revealed that *Actinobacteria*, *Alphaproteobacteria*, and *Acidobacteria* were among the main bacterial phyla. The main fungal phyla included *Pezizomycetes*, *Sordariomycetes*, *Dothideomycetes*, and *Agaricomycetes*. Nevertheless, the community compositions of

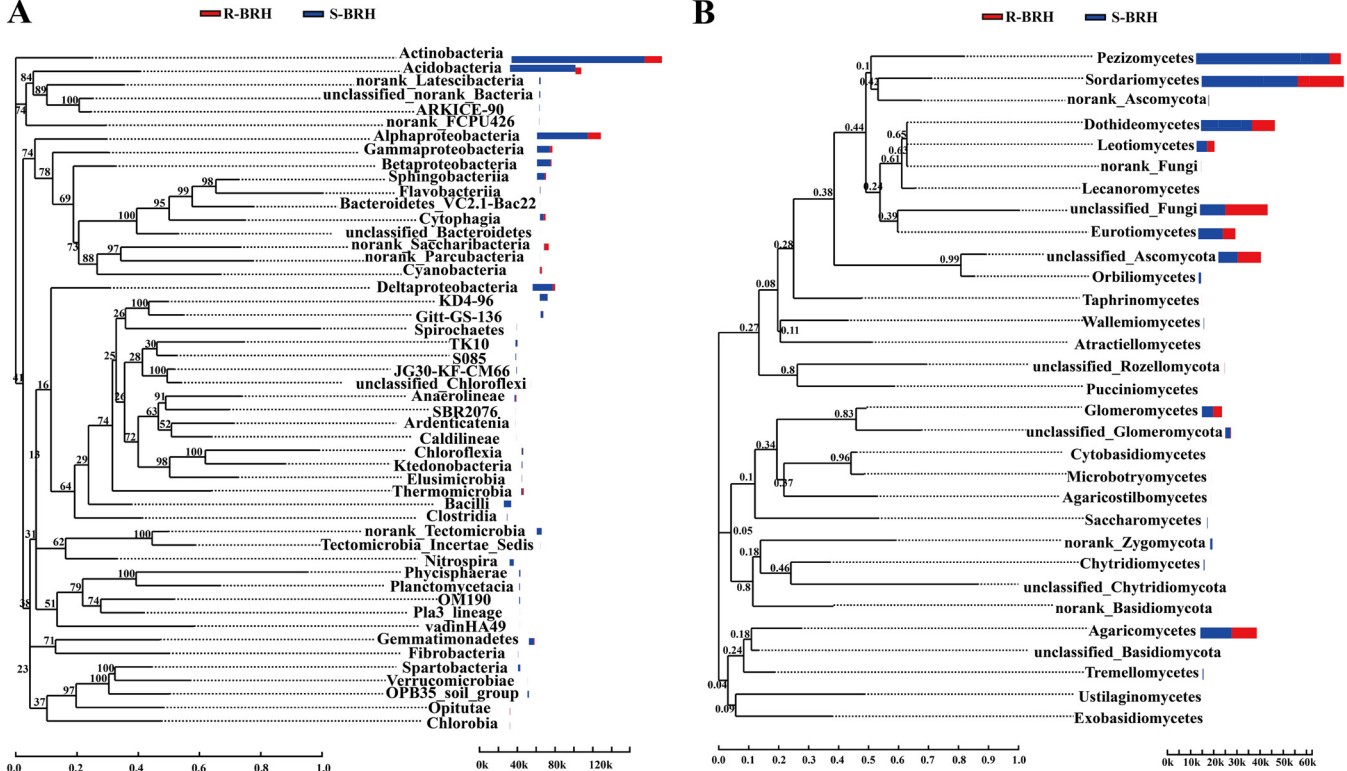

**FIG 5** Phylogenetic relationships and genera of the top 50 most abundant bacterial (A) and fungal (B) taxa. The phylogenetic evolutionary tree is on the left. Each branch in the evolutionary tree represents a species. The branches are colored according to the advanced taxonomic level to which the species belongs. The bar chart on the right shows the proportion of reads of species in different groups.

the soil and roots displayed significant variations for the fungal and bacterial communities. Furthermore, many of the core microbiomes exhibited a preference for certain habitats and had varied read proportions in the roots and soil (Fig. 5A). For the bacterial community, the proportions of *Saccharibacteria* and *Cyanobacteria* reads in the roots were higher than those in the soil, and no core microbiomes were found in the roots, which appeared in other cases. For the fungal community, the *Pezizomycetes*, *Dothideomycetes*, and *Sordariomycetes* were among the three most abundant taxa of fungi in soil on average (Fig. 5B).

**Environmental factors and community structure relationships.** To assess the taxonomic structures of the fungal and bacterial communities and their association with environmental conditions, a Pearson correlation heatmap was created (Fig. 6). The heatmap revealed that there were differences in the relationships between fungal and bacterial classes and environmental parameters. For the bacterial community, *Tectomicrobia* showed an extremely significant positive correlation with available potassium (AK). *Ktedonobacteria* revealed a significant positive correlation with available phosphorus (AP). ARKICE-90 and *Spartobacteria* showed significantly positive correlations with AP. TK10 had a strong positive association with organic matter (OM) and an even stronger but negative significant correlation with TP. The *Phycisphaerae* had an exceptionally strong positive connection with OM and a very strong negative and significant correlation with AP. The *Actinobacteria* showed a strong positive connection with ammonium nitrogen (AN) and a strong negative link with TN. The *Tectomicrobia* had an exceedingly strong positive connection with AN and a very strong negative and significant correlation with TN. The *Opitutae* had a very strong negative association with TN as well as a strong negative link with AN. The *Verrucomicrobiae* were significantly negatively correlated with TN (Fig. 6A). For the fungal community, *Tremellomycetes* and AP showed a highly significant and positive correlation. The *Chytridiomycetes* were also significantly positively correlated with pH (Fig. 6B).

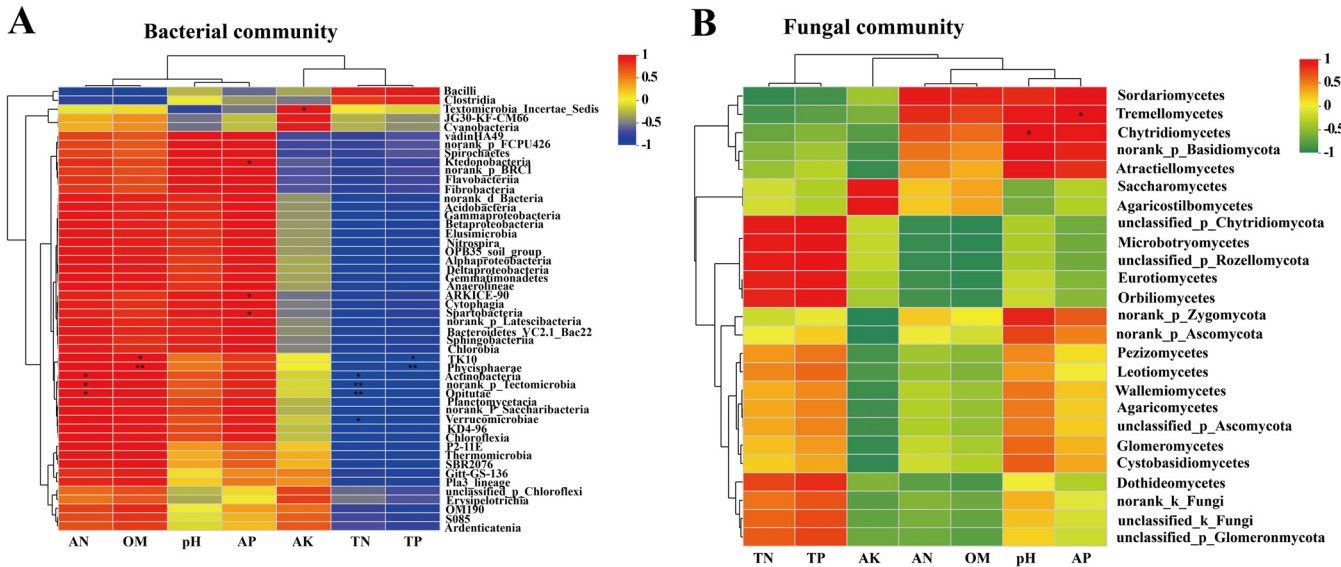

**FIG 6** Correlation heat map of the top 50 classes and soil properties of the communities of bacteria (A) and fungi (B). Environmental and class factors are represented by the *x* and *y* axes, respectively. The color range of different *R* values is shown in the right key. *P* values of <0.05 were considered significant and are indicated with an asterisk.

## DISCUSSION

**Function and structural diversity of *Helianthemum songaricum* soil microbial communities.** *Helianthemum songaricum* is a plant species that belongs to the old Mediterranean remnant, is confined to Central Asia, and is the dominant species in western Ordos, Inner Mongolia, People's Republic of China, which has a unique desert ecosystem (17). As a rare and endangered species in China, it has attracted widespread attention. To date, limited information is available about the microorganisms associated with the soil-root continuum of *Helianthemum songaricum*, although microbiome associations have been widely researched in other related endangered plants as well (18). In order to increase our knowledge about the root endophytic and rhizosphere microbial communities associated with this shrub, in this study, the community structure and diversity of microorganisms existing in the roots and soil of *Helianthemum songaricum* were researched. The analysis revealed that the sensitivity of communities in the endosphere was greater than that of communities in the soil rhizosphere. Furthermore, the bacterial communities in the root and soil environments also had higher sensitivities than the fungal communities. Compared with the rarefaction curves of fungi, the sequencing depth of bacteria showed more OTUs, and there was less variability between the rhizosphere and roots. This may be due to the high variability of ITS sequences, which cannot accurately identify all species (19, 20).

The alpha diversity of *Helianthemum songaricum* was comprehensively analyzed. The results indicated that the number of OTUs, the Chao1 index, and the Shannon index for the communities of microbes in *Helianthemum songaricum* soil were higher than those for the root endosphere (see Table S2 in the supplemental material). The number of OTUs in the bacterial soil samples (2,118) was much higher than that in the roots (314). This may be due to the interactions between plants and soil microorganisms where plants will provide some nutrients to soil microorganisms, which is conducive to the growth of the microorganisms. According to previous studies, the plant recruits the root-associated microbiome from the soil microbiome pool, which is then filtered through the rhizosphere. As a result, the density of microbes in the root endosphere was also lower than that in the rhizosphere (21, 22). The *Firmicutes*, *Bacillales*, *Gaiellales*, *Actinobacteria*, *Ascomycota*, *Sordariales*, and *Chaetomiaceae*, which are featured taxa isolated from rhizosphere soil samples, dominated the rhizosphere soil ecosystems, according to previous LDA effect size studies (23, 24). In terms of root endosphere communities, the main members of the microbial communities included *Alphaproteobacteria*, *Sphingomonadales*,

*Cytophagia*, *Saccharibacteria*, *Macrodidyma*, *Ilyonectria*, and *Acidobacteria* (Fig. 2). Previous reports have shown that root endophytic communities are dominated by the above-mentioned taxa. It has been hypothesized that these species have positive impacts on plant health and production, particularly in the areas of the suppression of pathogens, nutrient cycling, and abiotic stress tolerance (25–27).

The *Ascomycota* and *Bacillales*, both of which had an LDA score of >5.0, were found in the S-BRH sample and were marked with the most significant bacterial taxa, as shown in Fig. 3. These results were consistent with those of previous studies that found the above-mentioned taxa in truffle grounds and the rhizosphere of Xinjiang jujube (28–30). The LDA effect size, among samples, revealed highly significant variations in the bacterial relative abundances compared to those of fungal taxa, representing that the communities of bacteria had far higher levels of sensitivity in the root endosphere. This may be because the bacteria have unique living habits and are more susceptible to the influence of the external environment, leading to their relatively high sensitivity levels (31). Also, communities in the soil displayed a lower sensitivity level than the root communities of the endosphere, including fewer biomarkers. However, the results of this study are different from those of previous studies. Thus, we speculate that the different plant root exudates have important effects on microbial species in the root system or that there is the potential for a substantial shift in the proportion of some microorganisms present at different phases of plant development.

**Comparison of the bacterial and fungal communities of the soil-root continuum.** Phylogenetic analysis of the major taxa of microbial communities associated with *Helianthemum songaricum* using phylogenetic evolutionary trees indicates that the microbial community reads in the roots and soil are different (Fig. 5). In addition, the results showed that the *Pezizomycetes* and *Actinobacteria*, fungal and bacterial classes, respectively, were the most dominant classes in the microbial communities in the roots and soil. The *Helianthemum songaricum* growing environment represents a typical extreme environment characterized as arid and barren. The *Actinobacteria* have a characteristic filamentous form and have the potential ability to degrade and use organic matter, which helps them adapt to survive in an arid environment (32). A cooccurrence network is composed of various interspecific interactions in which species can be used to investigate correlations among microbial groups (33). The results of the network analysis showed that the soil has more nodes than the roots (Fig. 4). This can be attributed to the finding that more core soil microorganisms were enriched in the soil than in the root (Fig. 2), which may lead to an increase in the modularity of its networks. Previous studies demonstrated that stronger interactions between bacterial communities in the soil environment can improve the stability of microbial communities, help them to quickly respond to changing environments, and promote plant growth (34–36). Compared with bacteria, the structural stability of the fungal network in the rhizosphere and root system was low, which indicated that the structure of the fungal community was prone to change in an arid environment, resulting in the loss of fungal diversity. The *Acidobacteria* and *Sordariomycetes* were the most abundant taxa and were most closely related to other taxa, suggesting that these two taxa have a leading role in the *Helianthemum songaricum* microbiomes of the roots and soil. As reported in the literature, the *Acidobacteria* are widespread in natural environments, even in extreme environments, and they have specific driving roles and ecological functions in the ecosystem (37, 38). The *Ascomycota* include many classes, but the class *Sordariomycetes* is considered the largest of all of the classes and includes many important endophytes, plant pathogens, saprobes, epiphytes, fungicolous taxa, and lichenicolous or lichenized taxa (39). Numerous studies have shown that *Sordariomycetes* are beneficial microorganisms that produce a vast array of biologically active molecules that inhibit pathogens (40). Therefore, the *Sordariomycetes* should also be key organisms in the *Helianthemum songaricum* rhizosphere. In summary, the spatial structures of different environmental ecologies may create various niches for specific groups of microbes, and the patterns of the distribution of the main microorganisms varied among spatial structures.

Take as a whole, three bacterial and two fungal classes were considered host-specific microbes (Fig. S3). These data suggested that despite many differences in growth environments, host genetics, and plant development, the soil and roots of shrubs recruited and maintain many similar microbial taxa (41). These bacteria could play a vital part in the ecology of *Helianthemum songaricum* and other shrubs, and they also play a dominant role (42). A previous study also reported that both *Actinobacteria* and *Acidobacteria* were the main bacterial groups in the root and soil communities of shrub plants (43). Furthermore, the *Acidobacteria* can also degrade carbon sources and polysaccharides of plants and play an important role in the rapid flow of energy (44). The abundance of *Actinobacteria* in soil was relatively high, which indicates that it may have a relationship with *Helianthemum songaricum* soil, with low nutrients and an arid environment. Moreover, *Alphaproteobacteria* and *Gemmatimonadetes* not only play a key role in the nitrogen cycle but also produce coenzyme and cytochrome, which affect plant growth, thereby inducing plant systematic resistance to disease (45). For fungi, the main classes in the roots and soil were *Pezizomycetes*, *Sordariomycetes*, *Dothideomycetes*, and *Agaricomycetes*. Compared with the roots, the soil harbored a greater diversity of *Pezizomycetes*, consistent with a previous report (46). This may be caused by the scattered distribution and uneven colonization of bacteria in rhizosphere soil, which may lead to diversity differences. Furthermore, in *Helianthemum songaricum*, the fungal compositions and distribution patterns were quite distinct in different compartments. For example, *Ascomycota* were specifically identified in the root endosphere, while in the soil, they were very rarely detected. In the roots, the *Sordariomycetes* class was mostly enriched in the root zone and rhizosphere. Similar findings have been reported previously for wheat and tomato (47, 48). These common results indicate that different spatial structures may create different microbial groups, and whether the microbes can colonize the plant roots and soil depends on the plant type, the microecological environment, and the signaling molecules. In addition, the microbiomes of the soil-root continuum can enhance the uptake of plant nutrients and can have an important impact on the accumulation and production of plant secondary metabolites. On the contrary, the root exudates can affect the diversity of soil and root microbial communities. These characteristics of microbiomes associated with the soil-root continuum can effectively promote *Helianthemum songaricum*'s utilization of fertility resources under barren soil conditions.

Microorganisms of the rhizosphere and endosphere play an important role in plant growth and evolution (49, 50). Interestingly, the abundance of *Pezizomycetes* in the rhizosphere soil (31.95%) was significantly higher than that in the root system (5.7%), which may be due to the fact that *Pezizomycetes* are conducive to nutrient absorption by plants in arid and barren soil environments (51, 52). Therefore, in order to adapt to environmental changes, plants have attracted some microbial groups, providing them with a habitat conducive to plant growth.

**Effects of environmental factors on the microbial community of the soil-root continuum.** As we know, the composition of the microorganisms of the soil-root continuum is influenced by environmental factors, and microbes and plants have an interdependent association to adapt to various harsh environments (53–55). In our study, Pearson correlation heatmap analyses were performed to further assess the different factors of soil that affect the microorganism community (Fig. 6). The heatmap revealed that AK, AP, OM, AN, and TN had a significant effect on the bacterial community, and we propose that these factors play a vital part in promoting the stability and productivity of the bacterial community. In contrast to the bacterial community, the most important factors determining the composition of the fungal population were pH and AP. These results were consistent with previous research that demonstrated the relative importance of both TN and TP in this shrub. Previous research has shown that environmental conditions and microbial community interactions affect the structure of microbial communities in the soil as well as the spread of microbial communities (56, 57). As reported in the literature, AP has an important role in the soil microbial group, which not only is the main phosphorus source for microorganisms but also has a decisive role in the classification of microorganisms (58, 59).

In addition, TN, AK, and OM are indispensable nutrient elements in all organisms. Previous studies have shown that in arid environments, soil microorganisms initiate

nitrogen-increasing utilization strategies to ensure an adequate nitrogen supply. Thus, rhizosphere microorganisms provide a nitrogen-rich environment for plant roots (60, 61). In this study, our results were consistent with those of previous studies showing that nitrogen and organic material were the main factors affecting the diversity and composition of the microbial community. Nacke et al. (62) also found that soil pH was a major factor in soil bacterial communities. In our study, we found that the soil bacteria were significantly positively correlated with soil pH. Previous findings demonstrated that the soil pH level can result in micronutrient deficiencies and nutrient imbalances, thus enhancing the competition for nutrients between plants and microorganisms in the microenvironment (28). Thus, soil pH is an important limiting factor for microorganisms in the desert environment and the major driving element for ecosystem functioning in arid and barren environments (63). Many studies indicate that soil nutrients can affect the microbial community, which is supported by our results. So there are a number of soil factors (e.g., nutrient availability, phosphorus, and pH) that affect the root-soil-associated microbiomes. The differences in these key soil factors reflected the differences in the responses of the microbiome communities of shrub plants under arid conditions (64).

In short, the growth and survival of microorganisms in soil are affected by many environmental factors. Further studies are needed to link the observed changes in the root-associated microbiome of *Helianthemum songaricum* with soil functionality to develop its potential for soil ecosystem services in endangered plants.

**Conclusions and recommendations.** This study is the first to report the soil-root continuum environment and its factors driving variations in the diversity of fungal and bacterial communities in the natural environment of *Helianthemum songaricum*. The dominant bacterial classes were *Actinobacteria*, *Acidobacteria*, and *Alphaproteobacteria*, and the dominant fungal classes were *Sordariomycetes* and *Pezizomycetes*. At the taxonomic level and the individual-OTU level, there were differences in the abundances of bacteria and fungi in the root and rhizosphere of *Helianthemum songaricum*, indicating that the microbial community has a unique niche in plants and soil. Also, the microbiomes of the soil-root continuum of *Helianthemum songaricum* were significantly affected by the key potential drivers pH, organic matter, and ammonium nitrogen. *Helianthemum songaricum* is an ancient and endangered species. Therefore, this study provides a theoretical basis for the future research and protection of the biodiversity of endangered plants by studying the microbial community diversity of endangered desert plants. In the future, research on the relationships between plant communities should be increased to further clarify the impact of environmental and microbial factors on host plants, which will help us to better understand the changes of endangered plant communities.

## MATERIALS AND METHODS

**Site selection and study area description.** The present study was performed in the town of Etuoke Banner of Ordos City in Qipanjing (39°22′03″N, 107°01′48″E) (Fig. 7), Inner Mongolia Autonomous Region, People's Republic of China (65). The climate of Qipanjing is representative of the temperate continental climatic type. The average annual temperature is 9.8°C, the average annual precipitation is 222 mm, the soil types are brown and chestnut soils, and the vegetation is dominated by shrubs and herbs, e.g., *Helianthemum songaricum*, *Convolvulus tragacanthoides*, *Dracocephalum fruticulosum*, *Artemisia ordosica*, and *Atraphaxis frutescens* (see Table S1 in the supplemental material). The shrub *Helianthemum songaricum* is widespread and naturally found in Ordos City, with no apparent environmental gradients. To explore the influence of the extremely arid and barren environment on the root and soil microbial communities of *Helianthemum songaricum*, sampling sites in Qipanjing were selected using a systematic approach.

**Sample collection and analysis.** *Helianthemum songaricum* root samples were used in the present study. The soil (2 mm) surrounding the roots was taken. Furthermore, the root and soil samples are designated R-BRH and S-BRH, respectively. After sieving (<4 mm), three composite samples were made from a total of 10 soil and root samples by thorough mixing (*in situ*). Part of the soil was stored in a fridge at 4°C to be used for chemical analysis, and another part was placed at −80°C for storage until extraction and molecular analysis of DNA and microbes, respectively. Furthermore, chemical analysis was also performed on the bulk soil (Appendix S1).

**DNA extraction, PCR amplification, and sequencing.** Root and soil samples (0.5 g) were used for DNA extraction using a PowerSoil DNA isolation kit (MoBio Laboratories, Carlsbad, CA, USA), according to the manufacturer's instructions, in triplicate, and subsequently pooled. 16S rRNA gene sequencing was carried out using primers designed with V3 and V4 target regions, where V3 and V4 were approximately

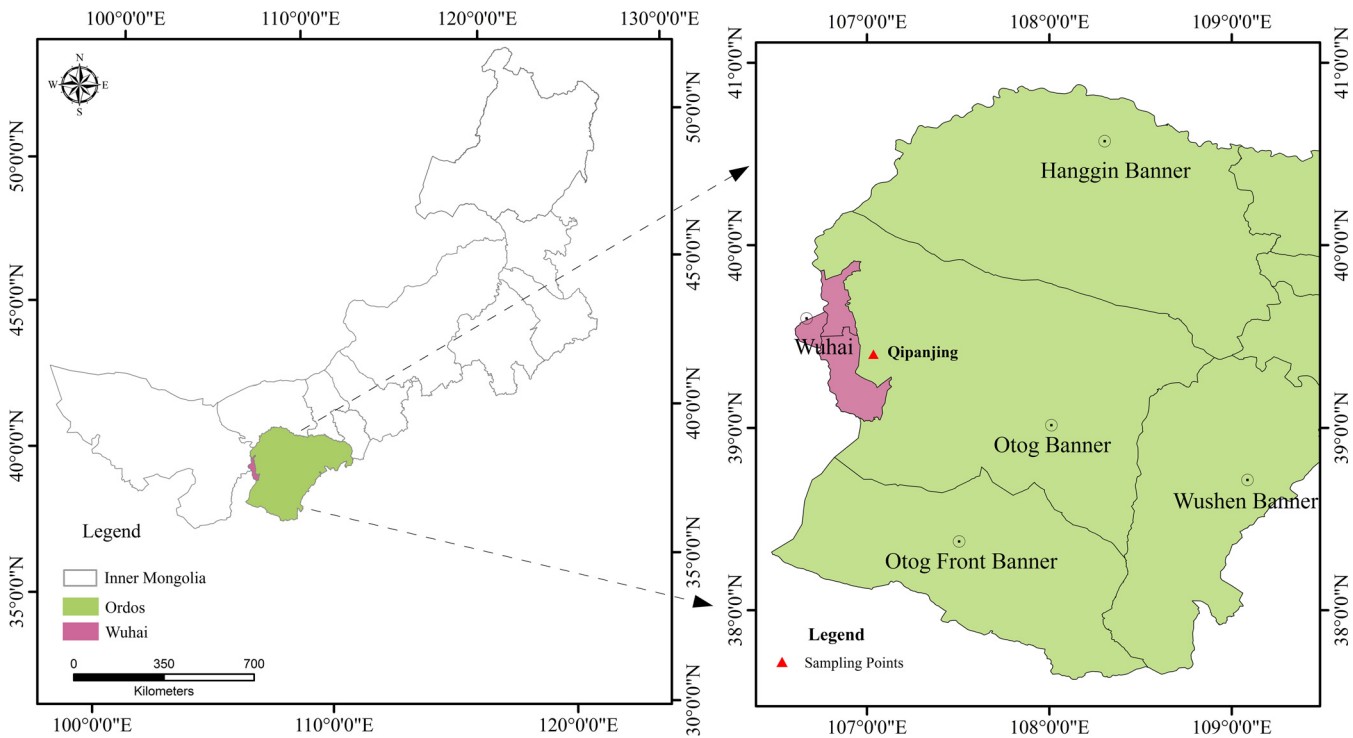

**FIG 7** Locations of the research sites. (Left) Inner Mongolia map. (Right) Sampling site locations.

469 bp (upstream primer 319F [5′-ACTCCTACGGGAGGCAGCAG-3′] and downstream primer 806R [5′-GGACTACHVGGGTWTCTAAT-3′]). Fungal ITS sequencing primers were designed with the ITS2 region as the target region (upstream primer fITS7 [5′-GTGARTCATCGAATCTTTG-3′] and downstream primer ITS4 [5′-TCCTCCGCTTATTGATATGC-3′]). To cover the whole bacterial and fungal communities, specific primers with barcodes targeting the amplified areas were synthesized and utilized for 454 pyrosequencing (66).

**Soil sampling and physicochemical properties.** Soil-to-water ratios of 1:2.5 and a Mettler Toledo benchtop pH meter were used to determine the soil pH. A soil analyzer (TPY-6pc; Zhejiang Top) was used to evaluate the soil available phosphorus (AP), potassium (AK), and ammonium nitrogen (AN) contents, whereas an elemental analyzer (Vario EL III) was used to calculate the organic matter (OM), total nitrogen (TN), and total phosphorus (TP) contents (67).

**Statistical analyses.** The statistical analyses in this study were performed using R 3.3.1. One-way analysis of variance (ANOVA) was used for data analysis (SPSS 25.0). Mega (version 10.0) (https://www.megasoftware.net/) analysis was used to produce the phylogenetic trees, and by selecting a sequence corresponding to a level of classification information, an evolutionary tree was constructed according to the ML (maximum likelihood) method and drawn using the R language (version 3.3.1).

For the detection of potential biomarkers at various taxonomic levels with a linear discriminant analysis (LDA) score of >4.0, the LDA effect size (LEfSe) method (http://huttenhower.sph.harvard.edu/galaxy/) was employed. NetWorkx was used to analyze species association networks and microbiome cooccurrence patterns (68). The heatmap displays the Pearson correlation coefficients of the top 50 most abundant bacterial and fungal classes and soil characteristics. ANOVA was used to perform correlation analyses, and Pearson coefficients with $P$ values of <0.05 were considered significant.

**Data availability.** Raw data have been uploaded and amplicon sequencing data have been deposited in the NCBI Sequence Read Archive (SRA) under accession numbers SRR22103821, SRR22103822, SRR22103823, SRR22103824, SRR22103825, and SRR22103826 for the 16 rRNA gene and SRR22116883, SRR22116884, SRR22116885, SRR22116886, SRR22116887, and SRR22116888 for the ITS.

## SUPPLEMENTAL MATERIAL

Supplemental material is available online only.
**SUPPLEMENTAL FILE 1**, PDF file, 1.9 MB.

## ACKNOWLEDGMENTS

This study was financially supported by the Inner Mongolia Natural Science Foundation Project (grant number 2015ZD04) and the National Natural Science Foundation of China (31760005).

Daolong Xu wrote the manuscript, with significant assistance and comments from Yuying Bao. Xiaowen Yu, Jin Chen, Haijing Liu, Yaxin Zheng, and Hanqing Qu performed the analysis with constructive discussions. All authors reviewed the manuscript.

We declare that there is no conflict of interest, i.e., neither personal nor any financial issues, that can affect the work reported or presented in this article.

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
