## [Reviewer comments · Microbiology Spectrum]

Microbiology Spectrum

Microbial assemblages associated with the soil-root continuum of endangered plants, *Helianthemum songaricum* Schrenk

Daolong Xu, Xiaowen Yu, Jin Chen, Haijing Liu, Yaxin Zheng, Hanting Qu, and Yuying Bao

Corresponding Author(s): Yuying Bao, Inner Mongolia University

Review Timeline:

Submission Date:	August 25, 2022
Editorial Decision:	October 13, 2022
Revision Received:	December 14, 2022
Editorial Decision:	March 6, 2023
Revision Received:	April 10, 2023
Accepted:	April 22, 2023

Editor: Zhongxiong Lai

Reviewer(s): Disclosure of reviewer identity is with reference to reviewer comments included in decision letter(s). The following individuals involved in review of your submission have agreed to reveal their identity: Gaosen Zhang (Reviewer #1); Madhusmita Borah (Reviewer #2)

Transaction Report:

DOI: <https://doi.org/10.1128/spectrum.03389-22>

October 13, 2022

Prof. Yuying Bao
Inner Mongolia University
Hohhot
China

Re: Spectrum03389-22 (**Microbial assemblages associated with the soil-root continuum of endangered plants, *Helianthemum songaricum* Schrenk**)

Dear Prof. Yuying Bao:

My decision is major revision. major concerns are: 1) data submission. Amplicon sequencing data need to be deposited. 2) poor writing.

Link Not Available

Sincerely,

Zhongxiong Lai

Journals Department
Reviewer comments:

Reviewer #1 (Comments for the Author):

Currently , the study is very primary, the description is unclear, and the conclusion regarding "sensitivities of community" is hard to be supported and even understand. If you want to show the uniqueness of this plant and its microbiota, comparisons with other plants, phylogenetically related or geologically neighbored, will be very helpful.

Reviewer #2 (Comments for the Author):

I would like to recommend the authors resubmit the manuscript after thorough and appropriate revisions in the texts.

Staff Comments:

Preparing Revision Guidelines

Please return the manuscript within 60 days; if you cannot complete the modification within this time period, please contact me. If you do not wish to modify the manuscript and prefer to submit it to another journal, please notify me of your decision immediately so that the manuscript may be formally withdrawn from consideration by Microbiology Spectrum.

Comments:

This paper reports an interesting work on root and soil microbiomes of *Helianthemum songaricum*. But I personally feel that the manuscript is written in poor English and needs proper revision. So, I would like to recommend the authors resubmit the manuscript after appropriate revisions, then only it will become suitable for publication in any reputed journal.

Revision Notes

Dear Editor:

Thank you for your kind letter of “Spectrum03389-22 - Editor decision - revise” on Oct 13, 2022.

Generally, we appreciate the editor and reviewer’s insightful comments, which are helpful for improving the manuscript. We are thankful to reviewer #1 and #2 comments, which are all valuable and very helpful for revising and improving our paper. Based on editor and reviewers' comments and requests, we have made major modification in the revised manuscript. Below is a summary of our responds to the reviewers' comments.

In revision notes, the line numbers refer to the PDF vision of the revised manuscript.

Thank you again for your comments. We hope we could learn more from you.

Editor insightful comments

1. Dada submission. Amplicon sequencing data need to be deposited.

We appreciate the editor insightful comments, and sorry for our negligence. We have listed the amplicon sequencing.

Line140-144: Raw data has been uploaded and amplicon sequencing data has been deposited. (16Sr DNA: SRR22103821 、 SRR22103822 、 SRR22103823
SRR22103824 、 SRR22103825 、 SRR22103826. ITS: SRR22116883 、
SRR22116884 、SRR22116885 、 SRR22116886 、 SRR22116887、SRR22116888)

2. Poor writing

We are thankful to the editor's insightful comments. This manuscript have already revised the grammatical errors by a native English speaker.

Thank you again for your comments. We hope we could learn more from you.

Reviewer's insightful comments

Thank you again for your comments. We hope we could learn more from you. We have highlighted the changes in red according to reviewer #1 comments in the annotated version of the revised manuscript (Revision, changes marked).

1. The study is very primary, the description is unclear, and the conclusion regarding "sensitivities of community" is hard to be supported and even understand.

We appreciate the reviewer's insightful comments. We are very sorry for not clearly describing this content in the previous manuscript. We have added and changed some informative and key sentences in the texts.

Line19-21: We have revised the sentence "We suspect that unknown microorganisms in roots and soil play an important role in the survival strategies of endangered plants".

Line21: "we investigated" instead of "we used Illumina MiSeq high-throughput sequencing to investigate.

Line22: Add a word "endangered".

Line23-24: We have revised the sentence "and observed that the microbial communities and structures between the rhizosphere and endosphere samples were distinguished".

Line25-38: We have revised the sentence "Rhizosphere bacteria were dominated by *Actinobacteria* (36.98%) and *Acidobacteria* (18.15%), whereas most endophytes were from the *Alphaproteobacteria* (23.17%) as well as *Actinobacteria* (29.94%). Rhizosphere bacteria relative abundance was higher than endosphere samples. Fungal rhizosphere and endophyte samples had approximately equal amounts of the *Sordariomycetes* (23%), while the *Pezizomycetes* were more abundant in the soil (31.95%) than root (5.70%). The abundance between root and soil samples of

phylogenetic relationships also showed that the most abundant bacterial and fungal reads tended to be dominant in either the soil or root samples but not both. Additionally, the Pearson correlation heatmap analysis showed that soil bacteria and fungal diversity and composition were closely related to pH, total nitrogen, total phosphorus and organic matter, of which pH, and organic matter were the main drivers. These results clarify the difference patterns of soil-root continuum of microbial communities in support of better conservation and utilization of desert endangered plants in Inner Mongolia".

Line 312-313: Changed "although root and soil microbiomes association have widely been researched in other related endangered plants as well" to "although microbiomes association have widely been researched in other related endangered plants as well".

Line 314-315: Changed "different microbiome communities" to "root endophytic and rhizosphere microbial communities".

Line 320-323: Changed "Additionally, the interactions between the fungal and bacterial communities; and soil parameters in the root and soil facilitated the development of intricate networks of species interactions" to "Compared with the rarefaction curves of fungi, the sequencing depth of bacteria was more OTUs, and the variability between rhizosphere and root was less. This may be due to the high variability of ITS sequence, which can not accurately identify all species (23,24)".

Line 327-330: Added "The number of OTU in the bacterial soil samples (2118) was much higher than in the root (314) . It may be due to the interaction between plants and soil microorganisms that plants will provide some nutrients to soil microorganisms, which is conducive to the growth of microorganisms.".

Line 359: Changed "Helianthemum songaricum soil-root continuum of bacterial and fungal communities differences" to "Comparison of the soil-root continuum of bacterial and fungal communities".

Line 377-380: Added "Compared with bacteria, the structure stability of fungal network in rhizosphere and root system was lower, which indicated that the structure of fungal community was prone to change in arid environment, resulting in the loss of fungal diversity.".

Line 395-396: Changed “in the root and soil the dominant bacterial class were Actinobacteria, Acidobacteria, Alphaproteobacteria, Deltaproteobacteria, Betaproteobacteria, Sphingobacteriia, Cytophagia and Gemmatimonadetes (Fig. 2).” to “three bacterial and two fungal class were considered as the host-specific microbes (Fig. S3)”

Line 412-413: Added “This may be caused by scattered distribution and uneven colonization of bacteria in rhizosphere soil, which may lead to diversity differences.”.

Line 428-434: Added “Rhizosphere and endosphere microorganisms play an important role in plant growth and evolution (53,54). Interestingly, the abundance of Pezizomycetes in rhizosphere soil (31.95%) is significantly higher than that in root system (5.7%), which may be due to the fact that Pezizomycetes is conducive to nutrient absorption by plants in arid and barren soil environment (55,56). Therefore, in order to adapt to environmental changes, plants have attracted some microbial groups, providing them with a habitat conducive to plant growth.”.

Line 435-436: Changed “Association between environmental factors and microbial community structure” to “Effects of environmental factors on the microbial community in the soil-root continuum”

Line 455-458: Added “Previous studies have shown that In arid environments, soil microorganisms initiate nitrogen-increasing utilization strategies to ensure nitrogen supply. Thus, rhizosphere microorganisms provide a nitrogen-rich environment for plant roots (65).”

Line 460: Added “Nacke et al.”

Line 462-467: Added “Previous findings have demonstrated that soil pH level can result in micronutrient deficiencies and nutrient imbalances, thus enhance the competition for nutrients between the plants and microorganisms in the microenvironment (67). So, soil pH is an important limiting factor for microorganisms in the desert environment, and the major driving element for ecosystem functioning in the arid and barren environment (68).”

Line 473-474: Added “the growth and survival of microorganisms in soil are affected by many environmental factors.”

Line 478-494: Changed “The present study reported first time, that the environmental and variation drivers of root-soil continuum rhizosphere of the fungal and bacterial communities diversity in the natural environment of *Helianthemum songaricum*. The bacterial community sensitivity was also greater than that of the community of fungi. Similarly, the sensitivity of the communities of roots endosphere was far more than rhizosphere soil communities. The bacterial community, while having a lower diversity of species, was more stable than the fungal community because it had a higher number of more integrated and complicated networks and biomarkers. We identified the core microbes and host-specific microbes, which increased the understanding of the soil-root continuum microbial communities of *Helianthemum songaricum*. Also, soil-root continuum microbiomes of *Helianthemum songaricum* were significantly affected by the key potential drivers of AP, AN, OM and pH. The findings in the present study will spawn new theoretical advancements and developments on the microbial community implications regarding the diversity in the adaptation of endangered plants to arid conditions.” to “This study was the first to report the soil-root continuum environment and its variation driving factors for the diversity of fungal and bacterial communities in the natural environment of *Helianthemum songaricum*. Bacterial dominant class were Actinobacteria, Acidobacteria and Alphaproteobacteria, and fungal dominant phyla were class were Sordariomycetes and Pezizomycetes. At the taxonomic level and individual OTU level, there are differences in the abundance of bacteria and fungi in the root and rhizosphere of *Helianthemum songaricum*, indicating that the microbial community has a unique niche in plants and soil. Also, soil-root continuum microbiomes of *Helianthemum songaricum* were significantly affected by the key potential drivers of pH, organic matter and ammonium nitrogen. *Helianthemum songaricum* is an ancient and endangered species. Therefore, this study provides a theoretical basis for the future research and protection of the endangered plant biodiversity by studying the microbial community diversity of desert endangered plants. In the future research, the relationship between plant communities should be increased to further clarify the

impact of environmental and microbial factors on host plants, which will help to better understand the changes of endangered plant communities.”.

2. If you want to show the uniqueness of this plant and its microbiota, comparisons with other plants, phylogenetically related or geologically neighbored, will be very helpful.

We appreciate the reviewer's insightful comments. This article describes the *Helianthemum songaricum Schrenk* plant separately because this plant is a member of the *Helianthemum songaricum Schrenk* family and the genus *Helianthemum songaricum Schrenk*, and its distribution area is very narrow, only in and around the West Ordos Nature Reserve in Inner Mongolia, China. In view of the small distribution area of *Helianthemum songaricum Schrenk*, we study its unique root system and inter-root microbial diversity to provide a reference for future conservation and transplanting using the biological properties of *Helianthemum songaricum Schrenk*. Thank you very much for the reviewer's comments. In the future further research, we will comparisons with other plants, phylogenetically related or geologically neighbored.

We have highlighted the changes in yellow according to reviewer #2 comments in the annotated version of the revised manuscript (Revision, changes marked).

1 I would like to recommend the authors resubmit the manuscript after thorough and appropriate revisions in the texts.

We appreciate the reviewer's insightful comments. According to the comments given by the reviewer, we have appropriate revisions in the texts.

Line43-49: We have revised the sentence "The symbiosis between soil microorganisms and these plants and their interaction with soil factors is an important feature of desert plant adaptation to arid and barren environment. Therefore, make a profound study on the microbial diversity of desert rare plants can provide important data support for the protection and utilization of desert rare plants. Accordingly, in this study, high-throughput sequencing technology was applied to study microbial diversity in the plant root and rhizosphere soils".

Line56: "high" instead of "a high", "multiple" instead of "a many multiple".

Line57-58: "plants" instead of "plant", "soils" instead of "soil".

Line59: "from" instead of "of".

Line61: "improve" instead of "increase", "avoid" instead of "reduce".

Line63: "can" instead of "may".

Line64: Add a phrase "the propagations of".

Line65-66: "can increase the adaptation of plants to extreme environments" instead of "may increase the capacity of applying these relationships to increase the adaptation of plants to extreme environments".

Line67-69: "the composition of microbial communities from endangered plants, including *Sarcozygium xanthoxylon*, *Tetraena mongolica*, and *Nitraria tangutorum* *Bobr* has been reported via high-throughput sequencing technology" instead of "many effective studies on sequencing technology (High-throughput) have been done to achieve the information about the composition of microbial communities for endangered plants, including *Sarcozygium xanthoxylon*, *Tetraena mongolica*, and *Nitraria tangutorum* *Bobr*".

Line70-71: "composition in the root and soil" instead of "dominant microbial".

Line71-72: "and the microbial species in the soil were more various than those in the endosphere" instead of "and the main microorganisms in the soil are much more various than in the endosphere".

Line73-75 : We have revised "the soil-root continuum microbiome of a typical endangered plant, *Helianthemum songaricum* Schrenk, has not been studied currently,

which, little is known about whether or not soil microbial variation is correlated with succession in these endangered plant".

Line76-78: "differences" instead of "more about the variations", "from" instead of "of", "growing in natural habitat" instead of "natural habitats", "the" instead of "their", "determined " instead of "examined".

Line79: "It has been known that" instead of "Many of the previous findings reported that".

Line79-80 : "exert" instead of "exerts", "beneficial influence on" instead of "exerts considerable influence over".

Line82-83: "including the improvement of disease and stress resistance capacity, and the acceleration of plant growth rate" instead of "including promoting disease resistance, plant growth, and stress resistance".

Line83-86: "previously" instead of "in the previous literature", "is" instead of "are", "the external environments, thus, exhibits complex diversity" instead of "the external environment and exhibits complex diversity", "structural feature" instead of "structure of the microbial community". "The soil-root continuum" instead of "in the previous literature"

Line 89: Added "changes".

Line 90: Changed "alter" to "promote".

Line 94: Changed "among soil-root continuum microbial community groups and different in situ remains unclear" to "different in situ remains unclear".

Line 98-104: Changed "To address these issues, the high-throughput sequencing (Illumina MiSeq) analyzed the fungal and bacterial communities of *Helianthemum songaricum* Schrenk (in Ordos City of Qipanjing). Also, we identified the core microbiomes and the enriched microbial. The soil-root microbial communities in dry environments are very complicated, and the present study findings will provide light on this complexity for endangered plants" to "To address these issues, we analyzed the fungal and bacterial communities from *Helianthemum songaricum* Schrenk growing in Ordos City of Qipanjing via high-throughput sequencing technology, enabling direct comparisons between these microbial communities. The main purpose

was to reveal the variation of soil-root microbial communities, elucidate the environmental factors that influence the root and soil microbial communities, and provide theoretical reference for the better conservation and utilization of desert endangered plants in Inner Mongolia”.

Line 140-144: Added “Raw data has been uploaded and amplicon sequencing data has been deposited. (16Sr DNA: SRR22103821 、 SRR22103822 、 SRR22103823 、 SRR22103824 、 SRR22103825 、 SRR22103826. ITS: SRR22116883 、 SRR22116884 、 SRR22116885 、 SRR22116886 、 SRR22116887 、 SRR22116888)”.

Line 149-150: Delete “and the only species of Cistaceae”.

Line 578-580: Added “23. Jumpponen A, Jones KL, Mattox D, and Yaege C. 2010. Massively parallel 454-sequencing of fungal communities in *Quercus* spp. ectomycorrhizas indicates seasonal dynamics in urban and rural sites. *Molecular Ecology*. 19: 41-53.”

Line 581-583: Added “24. Buee M, Reich M, Murat C, Morin E, Nilsson RH, Uroz S, Martin F. 2009. 454 Pyrosequencing analyses of forest soils reveal an unexpectedly high fungal diversity. *New Phytologist*. 184:449-456.”

Line 682-685: Added “53. Lu GH, Tang CY, Hua XM, Cheng J, Wang GH, Zhu YL, Zhang LY, Shou HX, Qi JL, Yang YH. 2018. Effects of an EPSPS-transgenic soybean line ZUTS31 on root-associated bacterial communities during field growth. *Plos One*, 13(2):e0192008.”

Line 686-689: Added “54. Youseif SH, Abd El-Megeed FH, Humm EA, Maymon M, Mohamed AH, Saleh SA, Hirsch AM. 2021. Comparative Analysis of the Cultured and Total Bacterial Community in the Wheat Rhizosphere Microbiome Using Culture-Dependent and Culture-Independent Approaches. *Microbiology spectrum*, 9(2): e00678-21.”

Line 690-692: Added “55. Li WJ, Li Y, Lv J, He XM, Wang JL, Teng DX, Jiang LM, Wang HF, Lv GH. 2021. Rhizosphere effect alters the soil microbiome composition and C, N transformation in an arid ecosystem. *Applied soil ecology*, 170,104296.”

Line 693-695: Added “56. Kearn J, McNary C, Lowman JS, Mei C, Aanderud ZT,

Smith ST, West J, Colton E, Hamson M, Nielsen BL. 2019. Salt-tolerant halophyte rhizosphere bacteria stimulate growth of alfalfa in salty soil. *Frontiers in Microbiology*. 10, 1849.”

Line 719-722: Added “65. Alejandra Miranda-Carrasco, Yendi E Navarro-Noya, Bram Govaerts, Nele Verhulst, Luc Dendooven. 2022. Nitrogen Fertilizer Application Alters the Root Endophyte Bacterial Microbiome in Maize Plants, but Not in the Stem or Rhizosphere Soil. *Microbiology spectrum*.”

Line 727-730: Added “67. Al-Sadi AM, Al-Khatri B, Nasehi A, Al-Shihi M, Al-Mahmooli IH, Maharachchikumbura SSN. 2017. High Fungal Diversity and Dominance by Ascomycota in Dam Reservoir Soils of Arid Climates. *International Journal of Agriculture and Biology*, 19(4), 682-688.”

Line 731-733: Added “68. Ouyang SN, Tian YQ, Liu QY, Zhang L, Wang RX, Xu XL, 2016. Nitrogen competition between three dominant plant species and microbes in a temperate grassland. *Plant Soil*, 408, 1-12.”

Thank you again for your comments. We hope we could learn more from you. Finally, we are very grateful to the reviewer’s and editor’s understanding and affirmation again. We wish the article can be published in Spectrum.

March 6, 2023

Prof. Yuying Bao
Inner Mongolia University
Hohhot
China

Re: Spectrum03389-22R1 (**Microbial assemblages associated with the soil-root continuum of endangered plants, *Helianthemum songaricum* Schrenk**)

Dear Prof. Yuying Bao:

Link Not Available

Sincerely,

Zhongxiong Lai

Journals Department
Reviewer comments:

Reviewer #3 (Comments for the Author):

I have reviewed the manuscript entitled "Microbial assemblages associated with the soil-root continuum of endangered plants, *Helianthemum songaricum* Schrenk". The authors compared bacterial and fungal community of the rhizosphere and endosphere of the *Helianthemum songaricum*. However, the English writing of the manuscript is very poor. It is very hard to follow. Suggest deep language editing.

Fig 3 and 5: the authors should provide error bar for each group.

Fig6: Suggest providing the p values and indicates correlation with $p < 0.05$.

Staff Comments:

Preparing Revision Guidelines

Please return the manuscript within 60 days; if you cannot complete the modification within this time period, please contact me. If you do not wish to modify the manuscript and prefer to submit it to another journal, please notify me of your decision immediately so that the manuscript may be formally withdrawn from consideration by Microbiology Spectrum.

Revision Notes

Dear Editor:

Thank you for your kind letter of “**Spectrum03389-22R1** - Editor decision - revise” on March 6, 2023.

Generally, we appreciate the editor and reviewer’s insightful comments, which are helpful for improving the manuscript. We are thankful to reviewer #1, #2 and #3 comments, which are all valuable and very helpful for revising and improving our paper. Based on editor and reviewers' comments and requests, we have made major modification in the revised manuscript. Below is a summary of our responds to the reviewers' comments.

In revision notes, the line numbers refer to the PDF vision of the revised manuscript.

Thank you again for your comments. We hope we could learn more from you.

Editor insightful comments

1. Dada submission. Amplicon sequencing data need to be deposited.

We appreciate the editor insightful comments, and sorry for our negligence. We have listed the amplicon sequencing.

Line140-144: Raw data has been uploaded and amplicon sequencing data has been deposited. (16Sr DNA: SRR22103821 、 SRR22103822 、 SRR22103823
SRR22103824 、 SRR22103825 、 SRR22103826. ITS: SRR22116883 、
SRR22116884 、SRR22116885 、 SRR22116886 、 SRR22116887、SRR22116888)

2. Poor writing

We are thankful to the editor's insightful comments. This manuscript have already revised the grammatical errors by a native English speaker.

Thank you again for your comments. We hope we could learn more from you.

Reviewer's insightful comments

Thank you again for your comments. We hope we could learn more from you. We have highlighted the changes in red according to reviewer #1 comments in the annotated version of the revised manuscript (Revision, changes marked).

1. The study is very primary, the description is unclear, and the conclusion regarding "sensitivities of community" is hard to be supported and even understand.

We appreciate the reviewer's insightful comments. We are very sorry for not clearly describing this content in the previous manuscript. We have added and changed some informative and key sentences in the texts.

Line19-21: We have revised the sentence "We suspect that unknown microorganisms in roots and soil play an important role in the survival strategies of endangered plants".

Line21: "we investigated" instead of "we used Illumina MiSeq high-throughput sequencing to investigate.

Line22: Add a word "endangered".

Line23-24: We have revised the sentence "and observed that the microbial communities and structures between the rhizosphere and endosphere samples were distinguished".

Line25-38: We have revised the sentence "Rhizosphere bacteria were dominated by *Actinobacteria* (36.98%) and *Acidobacteria* (18.15%), whereas most endophytes were from the *Alphaproteobacteria* (23.17%) as well as *Actinobacteria* (29.94%). Rhizosphere bacteria relative abundance was higher than endosphere samples. Fungal rhizosphere and endophyte samples had approximately equal amounts of the *Sordariomycetes* (23%), while the *Pezizomycetes* were more abundant in the soil (31.95%) than root (5.70%). The abundance between root and soil samples of

phylogenetic relationships also showed that the most abundant bacterial and fungal reads tended to be dominant in either the soil or root samples but not both. Additionally, the Pearson correlation heatmap analysis showed that soil bacteria and fungal diversity and composition were closely related to pH, total nitrogen, total phosphorus and organic matter, of which pH, and organic matter were the main drivers. These results clarify the difference patterns of soil-root continuum of microbial communities in support of better conservation and utilization of desert endangered plants in Inner Mongolia".

Line 312-313: Changed “although root and soil microbiomes association have widely been researched in other related endangered plants as well” to “although microbiomes association have widely been researched in other related endangered plants as well”.

Line 314-315: Changed “different microbiome communities” to “root endophytic and rhizosphere microbial communities”.

Line 320-323: Changed “Additionally, the interactions between the fungal and bacterial communities; and soil parameters in the root and soil facilitated the development of intricate networks of species interactions” to “Compared with the rarefaction curves of fungi, the sequencing depth of bacteria was more OTUs, and the variability between rhizosphere and root was less. This may be due to the high variability of ITS sequence, which can not accurately identify all species (23,24)”.

Line 327-330: Added “The number of OTU in the bacterial soil samples (2118) was much higher than in the root (314) . It may be due to the interaction between plants and soil microorganisms that plants will provide some nutrients to soil microorganisms, which is conducive to the growth of microorganisms.”.

Line 359: Changed “*Helianthemum songaricum* soil-root continuum of bacterial and fungal communities differences” to “Comparison of the soil-root continuum of bacterial and fungal communities”.

Line 377-380: Added “Compared with bacteria, the structure stability of fungal network in rhizosphere and root system was lower, which indicated that the structure of fungal community was prone to change in arid environment, resulting in the loss of fungal diversity.”.

Line 395-396: Changed “in the root and soil the dominant bacterial class were Actinobacteria, Acidobacteria, Alphaproteobacteria, Deltaproteobacteria, Betaproteobacteria, Sphingobacteriia, Cytophagia and Gemmatimonadetes (Fig. 2).” to “three bacterial and two fungal class were considered as the host-specific microbes (Fig. S3)”

Line 412-413: Added “This may be caused by scattered distribution and uneven colonization of bacteria in rhizosphere soil, which may lead to diversity differences.”.

Line 428-434: Added “Rhizosphere and endosphere microorganisms play an important role in plant growth and evolution (53,54). Interestingly, the abundance of Pezizomycetes in rhizosphere soil (31.95%) is significantly higher than that in root system (5.7%), which may be due to the fact that Pezizomycetes is conducive to nutrient absorption by plants in arid and barren soil environment (55,56). Therefore, in order to adapt to environmental changes, plants have attracted some microbial groups, providing them with a habitat conducive to plant growth.”.

Line 435-436: Changed “Association between environmental factors and microbial community structure” to “Effects of environmental factors on the microbial community in the soil-root continuum”

Line 455-458: Added “Previous studies have shown that In arid environments, soil microorganisms initiate nitrogen-increasing utilization strategies to ensure nitrogen supply. Thus, rhizosphere microorganisms provide a nitrogen-rich environment for plant roots (65).”

Line 460: Added “Nacke et al.”

Line 462-467: Added “Previous findings have demonstrated that soil pH level can result in micronutrient deficiencies and nutrient imbalances, thus enhance the competition for nutrients between the plants and microorganisms in the microenvironment (67). So, soil pH is an important limiting factor for microorganisms in the desert environment, and the major driving element for ecosystem functioning in the arid and barren environment (68).”

Line 473-474: Added “the growth and survival of microorganisms in soil are affected by many environmental factors.”

Line 478-494: Changed “The present study reported first time, that the environmental and variation drivers of root-soil continuum rhizosphere of the fungal and bacterial communities diversity in the natural environment of *Helianthemum songaricum*. The bacterial community sensitivity was also greater than that of the community of fungi. Similarly, the sensitivity of the communities of roots endosphere was far more than rhizosphere soil communities. The bacterial community, while having a lower diversity of species, was more stable than the fungal community because it had a higher number of more integrated and complicated networks and biomarkers. We identified the core microbes and host-specific microbes, which increased the understanding of the soil-root continuum microbial communities of *Helianthemum songaricum*. Also, soil-root continuum microbiomes of *Helianthemum songaricum* were significantly affected by the key potential drivers of AP, AN, OM and pH. The findings in the present study will spawn new theoretical advancements and developments on the microbial community implications regarding the diversity in the adaptation of endangered plants to arid conditions.” to “This study was the first to report the soil-root continuum environment and its variation driving factors for the diversity of fungal and bacterial communities in the natural environment of *Helianthemum songaricum*. Bacterial dominant class were Actinobacteria, Acidobacteria and Alphaproteobacteria, and fungal dominant phyla were class were Sordariomycetes and Pezizomycetes. At the taxonomic level and individual OTU level, there are differences in the abundance of bacteria and fungi in the root and rhizosphere of *Helianthemum songaricum*, indicating that the microbial community has a unique niche in plants and soil. Also, soil-root continuum microbiomes of *Helianthemum songaricum* were significantly affected by the key potential drivers of pH, organic matter and ammonium nitrogen. *Helianthemum songaricum* is an ancient and endangered species. Therefore, this study provides a theoretical basis for the future research and protection of the endangered plant biodiversity by studying the microbial community diversity of desert endangered plants. In the future research, the relationship between plant communities should be increased to further clarify the

impact of environmental and microbial factors on host plants, which will help to better understand the changes of endangered plant communities.”.

2. If you want to show the uniqueness of this plant and its microbiota, comparisons with other plants, phylogenetically related or geologically neighbored, will be very helpful.

We appreciate the reviewer's insightful comments. This article describes the *Helianthemum songaricum Schrenk* plant separately because this plant is a member of the *Helianthemum songaricum Schrenk* family and the genus *Helianthemum songaricum Schrenk*, and its distribution area is very narrow, only in and around the West Ordos Nature Reserve in Inner Mongolia, China. In view of the small distribution area of *Helianthemum songaricum Schrenk*, we study its unique root system and inter-root microbial diversity to provide a reference for future conservation and transplanting using the biological properties of *Helianthemum songaricum Schrenk*. Thank you very much for the reviewer's comments. In the future further research, we will comparisons with other plants, phylogenetically related or geologically neighbored.

We have highlighted the changes in yellow according to reviewer #2 comments in the annotated version of the revised manuscript (Revision, changes marked).

1. I would like to recommend the authors resubmit the manuscript after thorough and appropriate revisions in the texts.

We appreciate the reviewer's insightful comments. According to the comments given by the reviewer, we have appropriate revisions in the texts.

Line43-49: We have revised the sentence "The symbiosis between soil microorganisms and these plants and their interaction with soil factors is an important feature of desert plant adaptation to arid and barren environment. Therefore, make a profound study on the microbial diversity of desert rare plants can provide important data support for the protection and utilization of desert rare plants. Accordingly, in this study, high-throughput sequencing technology was applied to study microbial diversity in the plant root and rhizosphere soils".

Line56: "high" instead of "a high", "multiple" instead of "a many multiple".

Line57-58: "plants" instead of "plant", "soils" instead of "soil".

Line59: "from" instead of "of".

Line61: "improve" instead of "increase", "avoid" instead of "reduce".

Line63: "can" instead of "may".

Line64: Add a phrase "the propagations of".

Line65-66: "can increase the adaptation of plants to extreme environments" instead of "may increase the capacity of applying these relationships to increase the adaptation of plants to extreme environments".

Line67-69: "the composition of microbial communities from endangered plants, including *Sarcozygium xanthoxylon*, *Tetraena mongolica*, and *Nitraria tangutorum* Bobr has been reported via high-throughput sequencing technology" instead of "many effective studies on sequencing technology (High-throughput) have been done to achieve the information about the composition of microbial communities for endangered plants, including *Sarcozygium xanthoxylon*, *Tetraena mongolica*, and *Nitraria tangutorum* Bobr".

Line70-71: "composition in the root and soil" instead of "dominant microbial".

Line71-72: "and the microbial species in the soil were more various than those in the endosphere" instead of "and the main microorganisms in the soil are much more various than in the endosphere".

Line73-75 : We have revised "the soil-root continuum microbiome of a typical endangered plant, *Helianthemum songaricum* Schrenk, has not been studied currently,

which, little is known about whether or not soil microbial variation is correlated with succession in these endangered plant".

Line76-78: "differences" instead of "more about the variations", "from" instead of "of", "growing in natural habitat" instead of "natural habitats", "the" instead of "their", "determined " instead of "examined".

Line79: "It has been known that" instead of "Many of the previous findings reported that".

Line79-80 : "exert" instead of "exerts", "beneficial influence on" instead of "exerts considerable influence over".

Line82-83: "including the improvement of disease and stress resistance capacity, and the acceleration of plant growth rate" instead of "including promoting disease resistance, plant growth, and stress resistance".

Line83-86:"previously" instead of "in the previous literature", "is" instead of "are", "the external environments, thus, exhibits complex diversity" instead of "the external environment and exhibits complex diversity", "structural feature" instead of "structure of the microbial community". "The soil-root continuum" instead of "in the previous literature"

Line 89: Added "changes".

Line 90: Changed "alter" to "promote".

Line 94: Changed "among soil-root continuum microbial community groups and different in situ remains unclear" to "different in situ remains unclear".

Line 98-104: Changed "To address these issues, the high-throughput sequencing (Illumina MiSeq) analyzed the fungal and bacterial communities of *Helianthemum songaricum* Schrenk (in Ordos City of Qipanjing). Also, we identified the core microbiomes and the enriched microbial. The soil-root microbial communities in dry environments are very complicated, and the present study findings will provide light on this complexity for endangered plants" to "To address these issues, we analyzed the fungal and bacterial communities from *Helianthemum songaricum* Schrenk growing in Ordos City of Qipanjing via high-throughput sequencing technology, enabling direct comparisons between these microbial communities. The main purpose

was to reveal the variation of soil-root microbial communities, elucidate the environmental factors that influence the root and soil microbial communities, and provide theoretical reference for the better conservation and utilization of desert endangered plants in Inner Mongolia”.

Line 140-144: Added “Raw data has been uploaded and amplicon sequencing data has been deposited. (16Sr DNA: SRR22103821、SRR22103822、SRR22103823、SRR22103824、SRR22103825、SRR22103826. ITS: SRR22116883、SRR22116884、SRR22116885、SRR22116886、SRR22116887、SRR22116888)”.

Line 149-150: Delete “and the only species of Cistaceae”.

Line 578-580: Added “23. Jumpponen A, Jones KL, Mattox D, and Yaege C. 2010. Massively parallel 454-sequencing of fungal communities in *Quercus* spp. ectomycorrhizas indicates seasonal dynamics in urban and rural sites. *Molecular Ecology*. 19: 41-53.”

Line 581-583: Added “24. Buee M, Reich M, Murat C, Morin E, Nilsson RH, Uroz S, Martin F. 2009. 454 Pyrosequencing analyses of forest soils reveal an unexpectedly high fungal diversity. *New Phytologist*. 184:449-456.”

Line 682-685: Added “53. Lu GH, Tang CY, Hua XM, Cheng J, Wang GH, Zhu YL, Zhang LY, Shou HX, Qi JL, Yang YH. 2018. Effects of an EPSPS-transgenic soybean line ZUTS31 on root-associated bacterial communities during field growth. *Plos One*, 13(2):e0192008.”

Line 686-689: Added “54. Youseif SH, Abd El-Megeed FH, Humm EA, Maymon M, Mohamed AH, Saleh SA, Hirsch AM. 2021. Comparative Analysis of the Cultured and Total Bacterial Community in the Wheat Rhizosphere Microbiome Using Culture-Dependent and Culture-Independent Approaches. *Microbiology spectrum*, 9(2): e00678-21.”

Line 690-692: Added “55. Li WJ, Li Y, Lv J, He XM, Wang JL, Teng DX, Jiang LM, Wang HF, Lv GH. 2021. Rhizosphere effect alters the soil microbiome composition and C, N transformation in an arid ecosystem. *Applied soil ecology*, 170,104296.”

Line 693-695: Added “56. Kearn J, McNary C, Lowman JS, Mei C, Aanderud ZT,

Smith ST, West J, Colton E, Hamson M, Nielsen BL. 2019. Salt-tolerant halophyte rhizosphere bacteria stimulate growth of alfalfa in salty soil. *Frontiers in Microbiology*. 10, 1849.”

Line 719-722: Added “65. Alejandra Miranda-Carrasco, Yendi E Navarro-Noya, Bram Govaerts, Nele Verhulst, Luc Dendooven. 2022. Nitrogen Fertilizer Application Alters the Root Endophyte Bacterial Microbiome in Maize Plants, but Not in the Stem or Rhizosphere Soil. *Microbiology spectrum*.”

Line 727-730: Added “67. Al-Sadi AM, Al-Khatiri B, Nasehi A, Al-Shihi M, Al-Mahmooli IH, Maharachchikumbura SSN. 2017. High Fungal Diversity and Dominance by Ascomycota in Dam Reservoir Soils of Arid Climates. *International Journal of Agriculture and Biology*, 19(4), 682-688.”

Line 731-733: Added “68. Ouyang SN, Tian YQ, Liu QY, Zhang L, Wang RX, Xu XL, 2016. Nitrogen competition between three dominant plant species and microbes in a temperate grassland. *Plant Soil*, 408, 1-12.”

Thank you again for your comments. We hope we could learn more from you. We have highlighted the changes in green according to reviewer #3 comments in the annotated version of the revised manuscript (Revision, changes marked).

1. Suggest deep language editing.

We are thankful to the editor’s insightful comments. This manuscript have already revised the grammatical errors by a native English speaker.

2. Fig 3 and 5: the authors should provide error bar for each group.

We appreciate the reviewer’s insightful comments. According to the comments given by the reviewer, we have appropriate revisions in the texts.

Fig 3: Different color nodes represent the microbial groups that are significantly

enriched in the corresponding group and have significant influence on the difference between groups. The light yellow nodes represent microbial groups that have no significant difference between groups, or have no significant effect on differences between groups. Circles indicate phylogenetic levels from Phylum to species. The diameter of each circle is proportional to the abundance of the group.

Fig 5: The co-occurrence network diagram visually displays the co-occurrence relationship of species in different samples. Nodes in the network represent sample nodes or species nodes, and the lines between sample nodes and species nodes represent the species contained in the samples. Species with abundance (number of sequences) greater than 50 are displayed by default.

3. Fig6: Suggest providing the p values and indicates correlation with $p < 0.05$.

We appreciate the reviewer's insightful comments. According to the comments given by the reviewer, we have appropriate revisions in the texts.

Note: The phylogenetic evolutionary tree is on the left. Each branch in the evolutionary tree represents a species. The branches are colored according to the advanced taxonomic level to which the species belongs. The bar chart on the right shows the proportion of Reads of species in different groups.

Thank you again for your comments. We hope we could learn more from you. Finally, we are very grateful to the reviewer's and editor's understanding and affirmation again. We wish the article can be published in Spectrum.

April 22, 2023

Prof. Yuying Bao
Inner Mongolia University
Hohhot
China

Re: Spectrum03389-22R2 (**Microbial assemblages associated with the soil-root continuum of endangered plants, *Helianthemum songaricum* Schrenk**)

Dear Prof. Yuying Bao:

Your manuscript has been accepted, and I am forwarding it to the ASM Journals Department for publication. You will be notified when your proofs are ready to be viewed.

Sincerely,

Zhongxiong Lai
Editor, Microbiology Spectrum
